# Facial Composite Generation with Iterative Human Feedback

Florian Strohm                           FLORIAN.STROHM@VIS.UNI-STUTTGART.DE
Ekta Sood                                     EKTA.SOOD@VIS.UNI-STUTTGART.DE
Dominike Thomas                    DOMINIKE.THOMAS@VIS.UNI-STUTTGART.DE
Mihai Bâce                                   MIHAI.BACE@VIS.UNI-STUTTGART.DE
Andreas Bulling                      ANDREAS.BULLING@VIS.UNI-STUTTGART.DE
*Institute for Visualisation and Interactive Systems, University of Stuttgart, Germany*

**Editor:** Editor's name

## Abstract

We propose the first method in which human and AI collaborate to iteratively reconstruct the human's mental image of another person's face only from their eye gaze. Current tools for generating digital human faces involve a tedious and time-consuming manual design process. While gaze-based mental image reconstruction represents a promising alternative, previous methods still assumed prior knowledge about the target face, thereby severely limiting their practical usefulness. The key novelty of our method is a collaborative, iterative query engine: Based on the user's gaze behaviour in each iteration, our method predicts which images to show to the user in the next iteration. Results from two human studies (N=12 and N=22) show that our method can visually reconstruct digital faces that are more similar to the mental image, and is more usable compared to other methods. As such, our findings point at the significant potential of human-AI collaboration for reconstructing mental images, potentially also beyond faces, and of human gaze as a rich source of information and a powerful mediator in said collaboration.

**Keywords:** gaze, mental image reconstruction, human-ai collaboration, interactive system

## 1. Introduction

Methods and tools to generate human faces have broad applicability in computer graphics, visual design, human-computer interaction (HCI), and beyond. For example, creating facial composites or photofits – visual reconstructions of a human face from someone's mind – is widely used in criminal investigations to reconstruct the appearance of a wanted person (Frowd et al., 2004, 2005; George et al., 2008). Another popular application domain is gaming in which users frequently want to generate digital avatars that are visually similar to them as quickly and effortlessly as possible. Previous work has explored two main approaches to achieve this goal: One approach involves methods to automatically convert a real face image into an avatar using image processing techniques (Kim et al., 2019; Hu et al., 2017). However, these methods require the target face to be visually available in advance, which is not always possible in particular for facial composites or photofits. The second and more common approach is to rely on software tools that, while allowing for fine-grained control of the generated images (Schwind et al., 2017), require a lot of tedious and time-consuming manual work, and are usually geared to a particular application domain.

A promising alternative are computational methods that directly reconstruct mental images from brain activity recorded while users look at carefully crafted visual stimuli

(Beliy et al., 2019; Güçlütürk et al., 2017; Shen et al., 2019; VanRullen and Reddy, 2019; Date et al., 2019; Shatek et al., 2019; Lin et al., 2019). However, measuring brain activity is impractical for most real-world systems given that it requires expensive and special-purpose equipment as well as extensive operator training. In contrast, human gaze can be measured using off-the-shelf hardware that has recently become both significantly more affordable and usable also by non-experts (Kassner et al., 2014). Given these advances, a recent line of work has started to investigate gaze-based mental image retrieval (Wang et al., 2019) as well as visual reconstruction (Sattar et al., 2017, 2020). While the former uses eye gaze features to retrieve a mental image from an existing database, gaze-based mental image reconstruction is profoundly more challenging given that the target image exists only in a user's mind.

Strohm et al. (2021) recently demonstrated the first method to reconstruct a mental image solely from eye gaze fixations. While their face reconstructions looked promising visually, their method still required the target face to be known in advance, which renders the method impractical for real applications. With the goal to address this fundamental limitation, we propose a novel method for gaze-based mental image reconstruction that does not require any prior knowledge about the target image. To achieve this goal, our iterative method implements the principle of human-AI collaboration (Akata et al., 2020) (see Figure 1): On the AI side, we propose a query engine that predicts the most relevant set of face image features for the next iteration while taking prior iterations into account. These features are then decoded into images using a decoder network and presented to the human. The human user looks at these images, searching for similar features to their mental image while their gaze behaviour is being recorded using an eye tracker. Subsequently, new face image features are extracted using the gaze-guided extractor proposed by Strohm et al. (2021). These steps are repeated multiple times with gaze-guided image features being extracted in each iteration. Finally, all extracted image features are combined into a single feature vector that is decoded into the final facial composite.

The contributions of our work are two-fold: First, we propose a novel gaze-based collaborative method for mental image reconstruction (GBC-MIR). In stark contrast to prior work, our method does not require prior knowledge about the target face but only a pre-trained gaze-guided image feature extractor. Furthermore, our method is domain independent – while we demonstrate an application for face image reconstruction, our method can be applied for any mental image reconstruction task. Second, we evaluate our method in a 12-participant user study and show that it outperforms the current state of the art in reconstruction quality while, at the same time, achieving higher usability.

## 2. Related work

**Mental image retrieval and reconstruction.** In mental image retrieval, a query engine typically presents selected, existing images to a user who then manually chooses the closest fit (Fang et al., 2021; Fang and Geman, 2005; Fang et al., 2020; Fang and Yuan, 2018; Fang et al., 2005). Based on this choice, the system then presents a new set of images and the user input process continues iteratively until the target image has been found. Instead of selecting existing images from a database, target images can also be created synthetically. In mental image reconstruction, synthetic images are refined in every round of user feedback,

until the user is satisfied with the result. Bontrager et al. (2018) integrated user feedback in such a fashion and used deep interactive evolution to reconstruct mental images, while Xu et al. (2019) improved their method by allowing control over specific facial features through relevance feedback. By choosing only pictures with appropriate features, the user allows these features to contribute to the next iteration of pictures. Zaltron et al. (2020) proposed Composite Generating Generative Adversarial Networks (CG-GAN) allowing the user to traverse the latent space of a pre-trained GAN. Users were able to select proposed faces to combine their information through a process called mutation. However, these methods rely on explicit user feedback rather than implicit behavioural cues.

**Gaze-based mental image retrieval and reconstruction.** Human gaze has recently attracted increasing research interests as a promising implicit feedback modality for mental image retrieval and reconstruction. Leading these efforts was work by Sattar et al. (2015) who used bags of visual words to address the task in an open-world setting. Stauden et al. (2018) further improved their feature extraction component by adding a pre-trained CNN, while Barz et al. (2020) improved the encoding of fixation sequences with a pre-trained SegNet and used a SVM for the final prediction task. In later work, Sattar et al. (2020) went beyond predicting only the target instance by also predicting the target class and attributes, while Wang et al. (2019) explored a setting in which gaze behaviour used for the target prediction was collected after showing a stimulus. Finally, Strohm et al. (2021) were the first to demonstrate the feasibility of reconstructing mental images using gaze behaviour only. Their method consisted of an encoder to extract image features and activation maps as well as a scoring network to compare human gaze maps with neural activation maps to predict a relevancy score for each extracted feature. These scores were used to combine features from multiple images into a single feature vector that was finally decoded into the mental image. While achieving promising results, their method requires prior knowledge severely limiting the usability of their method.

## 3. Gaze-based collaborative mental image reconstruction (GBC-MIR)

The computational task that we study is that of visually reconstructing mental images from eye gaze fixations without prior knowledge of the target. We approach this by showing multiple, generated auxiliary images to a user and recording their gaze while they look at these faces. The auxiliary images are encoded and relevant image features are extracted based on the user's gaze behaviour. The extracted features can subsequently be decoded to reconstruct the mental image. Strohm et al. (2021) formulated this task as a mapping $\{(I_i, G_i)|i = 1...n\} \mapsto I_M$ , where $I_i$ are the auxiliary images shown to the user, $G_i$ are eye gaze fixations of that user on these images, and $I_M$ is the to-be-reconstructed mental (target) image that the user has in mind. The fundamental difficulty of this task is to extract sufficient information about the mental image from the fixations on the given set of auxiliary images. They used a small set of six auxiliary images, generated using prior knowledge. This ensured that enough information about the target could be extracted from the gaze data. However, given that information about the target image is not available for most real world use cases (e.g., facial composites in law enforcement), this renders their method impractical for actual use.

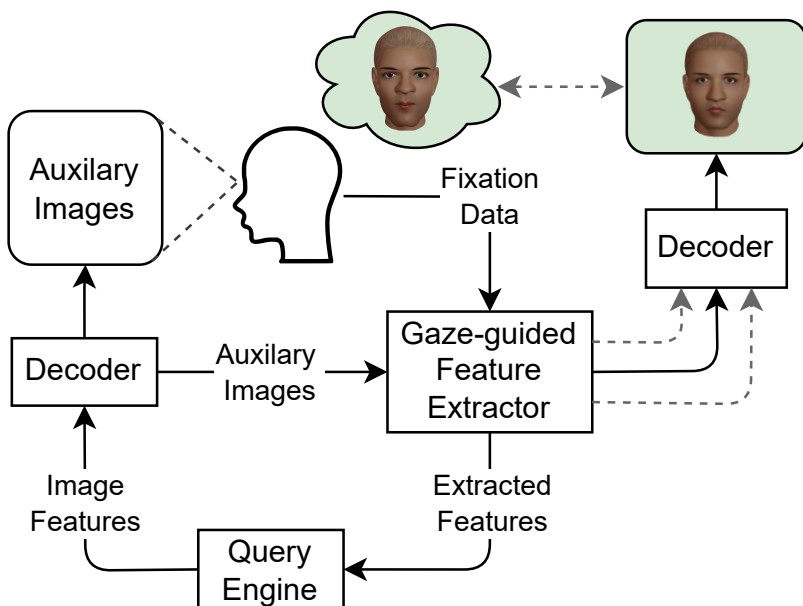

Figure 1: Overview of our method for gaze-based collaborative mental image reconstruction (GBC-MIR). With a specific mental image in mind, users observe auxiliary images proposed by a *query engine* and visually reconstructed by a *decoder*. A *gaze-guided feature extractor* extracts image features that are relevant to the mental image. Based on these features, the query engine proposes image features for the next iteration. This loop continues for several iterations until all extracted feature vectors are combined to reconstruct the final mental image.

To address this fundamental limitation, we present a collaborative human-AI system for mental image reconstruction that does not require any prior knowledge of the target image. Instead, our method iteratively reconstructs the target image by presenting sets of images to the user, thus performing a mapping $\{(I_{i,j}, G_{i,j})|i = 1...n\} \mapsto F_{M,j}, \ j = 1...m$, where $F_{M,j}$ represents extracted gaze-guided image features from iteration $j \in [1, 2, ..., m]$. After $m$ iterations, in which our query engine dynamically selects image features based on the user's gaze, our system combines the extracted features of each round to generate the target image. The overall architecture of our system is shown in Figure 1 and its core components are described in the following.

**Gaze-guided feature extractor (GFE).** A central component of our collaborative system is the gaze-guided feature extractor (GFE). It is represented by a pre-trained model that implements the function $\{(I_i, G_i)|i = 1...n\} \mapsto F$, i.e., given a set of auxiliary images $I$, it extracts a single feature vector $F$ based on the user's gaze fixations $G$. The extractor decides, for each feature dimension $f \in F$, from which auxiliary image $I_i$ to select a feature. That is, $F$ is composed of features from the set of auxiliary images $I$.

**Decoder.** The decoder maps feature vectors $F$ into the image domain $I$ using a pre-trained model. This allows us to visualise features for the user (in the form of faces) and to collect gaze data, which in turn can be used to select relevant features through the GFE. It is crucial that both the decoder and the GFE operate in the same feature space $\mathbb{F}$ to allow for an iterative loop of visually decoding features for users to look at, and encoding the images again using the recorded gaze information.

**Query engine.** For our system to maximise the information about the user's mental image, it needs to decide which images to show in each iteration. For this, we propose a novel query engine which predicts the image features $F_{i,j+1}$ for the next iteration, given the previously shown and extracted features, $F_{i,j}$ and $F_{M,j}$:

$$P(F_{i,j+1}|F_{i,j}, F_{M,j}), \text{ with } F_{i,0} \text{ and } F_{M,0} = \text{constant} \tag{1}$$

This allows our system to dynamically show images to the user based on their previous gaze behaviour. Thus, in each iteration, it can dynamically decide for which features to collect more information.

**Iterative collaboration.** Together, the GFE, decoder, and query engine form an interactive feedback loop, enabling the human and the system to collaboratively reconstruct the mental image as shown in Figure 1: The query engine proposes a set of feature vectors about which our system wants to gain knowledge, which are visually decoded into the image space by the decoder. Our system then shows these images to the user while their gaze fixations are being recorded. Using the joint gaze and image information, the GFE then extracts the feature vector that is most relevant in this iteration for eventually reconstructing the mental image. This cycle continues for $m$ iterations after which all the feature vectors extracted by the GFE are combined and visually decoded into the final mental image.

---

**Algorithm 1:** Selection layer simulating the gaze-guided feature extractor

---

**Input:** Auxiliary and target image features $f_{I,j}$ and $f_M$. Probability distribution $P_{\text{GFE}}$
**Output:** Selected features $f_{M,j}$
differences $\leftarrow$ abs($f_{I,j} - f_M$);
args $\leftarrow$ arg_sort(differences);
$k \sim P_{\text{GFE}}$;
selected_arg $\leftarrow$ one_hot(args[k]);
$f_{M,j} \leftarrow$ (selected_arg $* f_{I,j}$);

---

### 3.1. Simulating the gaze-guided feature extractor

Training this system end-to-end is impractical given that human gaze data for the GFE would have to be collected dynamically during training for each set of images predicted by the query engine. A key innovation of our method is that we instead simulate the GFE in the form of a selection layer during training, and only use a pre-trained GFE model, which requires gaze data, at test time.

The GFE is simulated as a probability distribution over the possible features that it can select from the auxiliary images. Given the user's fixation data, the GFE predicts, for each

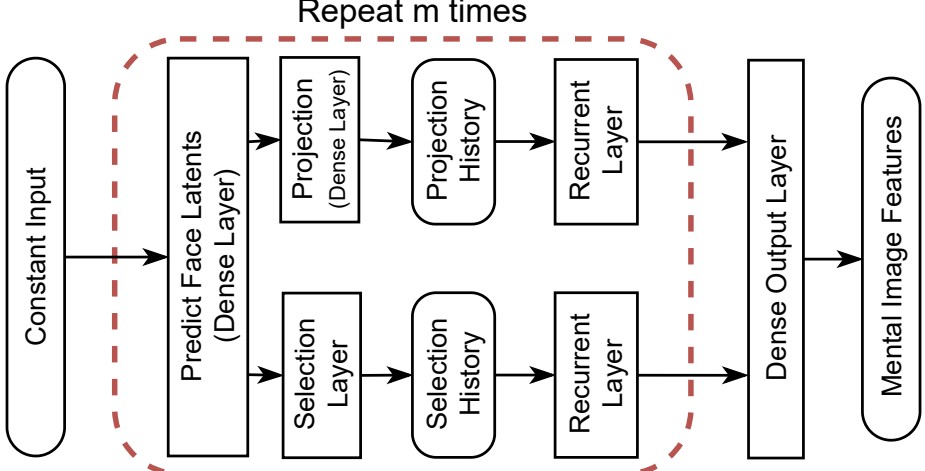

Figure 2: Overview of our end-to-end neural implementation of the collaborative reconstruction system. Dense layers predict the auxiliary image features for the next iteration. For this, they use the input of two recurrent layers that combine information about the shown and selected features of previous iterations. A final dense output layer combines the information of all selected feature vectors to predict the mental image features.

feature dimension $f \in F$, from which of the shown auxiliary images $I$ to select the value of $f$. Using a small amount of labelled data $\{(I_i, G_i, I_M)|i = 1...n\}$ we can estimate how often the pre-trained GFE selects the feature value closest to the target, the second closest, and so on. Specifically, we estimate $P_{\text{GFE}}(f_M = f_k)$, where $k$ represents the similarity rank of $f_k$ to the mental image feature $f_M$ (see Appendix A for details). Using these estimated probabilities we define a selection layer in Algorithm 1.

Using the target feature value $f_M$, we calculate a distance ranking of the proposed feature values $f_{I,j} = (f_{1,j}, f_{2,j}, ..., f_{n,j})$, $f_{i,j} \in F_{i,j}$ for iteration $j$. The layer selects the $k$-th best feature value $f_I$ by sampling $k$ from $P_{\text{GFE}}(f_M = f_k)$. Instead of directly indexing the selected value from the auxiliary image, we construct a one-hot selection vector and calculate the dot product with the features of the auxiliary images $f_{I,j}$. This way we model the selection as an external random node, allowing us to calculate gradients for each parameter in the neural network.

The GFE performs the mapping $\{(I_i, G_i)|i = 1...n\} \mapsto F$, while the selection layer implements the mapping $\{F_i|i = 1...n, F_M\} \mapsto F$. This approach allows us to predict the extracted features $F$ of each iteration without gaze data but requires the target image features $F_M$ as an input. Since the target face and the corresponding features are known at training time, this proxy mapping allows us to train GBC-MIR end-to-end.

## 3.2. End-to-end Training

The only component that has to be learned is the query engine since pre-trained models are used for the GFE and decoder. To train the query engine end-to-end we combine it with the selection layer defined in Algorithm 1 to create the GBC-MIR feedback loop. Figure 2 shows the neural network structure used to learn a set of recurrent and dense layers forming the query engine. The network is composed of $m$ stacked iteration modules as indicated by the dotted red line in Figure 2. Each module consists of a dense layer predicting $n$ auxiliary image feature vectors for that iteration based on the output of the previous module, which are fed into two separate network paths: One path starts with a selection layer that simulates the pre-trained GFE and receives $n * |F|$ features as input. It outputs the selected features $F_{j,M}$ of iteration $j$ based on Algorithm 1. These features are input to a recurrent layer together with the history of features from all previous iterations $\{(F_{M,k})|k = j - 1, j - 2, ...1\}$, combining the vector sequence into a single feature vector. The second path starts with a dense projection layer which also receives $n * |F|$ features as an input and outputs a lower-dimensional representation of these features. Together with the history of projected features from previous iterations, these are input into another recurrent layer combining the features. The projection path enables the network to keep track of which features it already proposed in any prior iteration. The outputs of each recurrent layer from the two paths, selection and projection, are concatenated and input to the next module of iteration $j + 1$. After $m$ iteration modules, the output of the last module is fed into a dense output layer which predicts the target image features.

The network is trained by minimising the mean squared error between predicted and target image features. While the network's input is only a constant value, it receives information about the target through the selection layer in each iteration module. At test time the selection layer is replaced with a pre-trained GFE, allowing the system to make user-specific predictions based on their gaze behaviour. In this case a pre-trained decoder generates the auxiliary images for each iteration given the predicted image features of the query engine. These images can then be shown to the user while recording their gaze behaviour. The generated images of the first iteration are constant, as our system has no knowledge about the mental image yet. However, subsequent predictions are made based on the previously extracted features of the GFE, which are user-specific and encode increasingly more information about their mental image.

## 4. Reconstruction experiments

For evaluation, we used the pre-trained GFE and decoder provided by Strohm et al. (2021) and thus evaluate GBC-MIR in the FaceMaker face image domain (Schwind et al., 2017). FaceMaker is an interactive tool to manually generate human-like faces by manipulating a set of 28 face appearance sliders, giving fine grained control over different facial features. We compare GBC-MIR with a baseline method and a FaceMaker control.

### 4.1. Methods

**Ours – Gaze-based collaborative mental image reconstruction (GBC-MIR).** To train our system we estimated the values $P_{\text{GFE}}$ of the pre-trained GFE model for the

selection layer (see appendix). As we have to reconstruct 28 features and the used GFE requires six input images, the dense layers of our system depicted in Figure 2 consist of $28 * 6 = 168$ neurons with a Sigmoid activation. The projection layers consist of 50 neurons with a ReLU activation, projecting the 168 image features into a 50 dimensional space. The recurrent layers are implemented by gated recurrent unit (GRU) (Cho et al., 2014) layers, each extracting 50 features from the history of selected and projected features, respectively. We set the number of iterations to $m = 10$, resulting in ten stacked iteration modules. The output dense layer consist of 28 neurons with Sigmoid activation.

The model was trained for 50 epochs on a V100 GPU with a batch size of 32, where each batch consists of randomly generated target feature vectors of size 28. We used the Adam optimiser (Kingma and Ba, 2014) with a learning rate of 0.0004 and default parameters otherwise. For the user study, we replaced the selection layer with the pre-trained GFE.

As in Strohm et al. (2021), the task of the participants in each iteration was to rank the six auxiliary images according to the similarity with their mental image. They had 30 seconds to complete this task after which the gaze data was used to extract features with the GFE. Since the used GFE requires 30 seconds of gaze recordings to extract image features, and we used ten iterations, the resulting total reconstruction time amounted to five minutes. This keeps the interaction time with the system short, reducing possible eye strain and ensuring fast reconstruction times.

**Baseline – Interactive evolution.** As the baseline we compared to an adapted version of the state-of-the-art mental image reconstruction method proposed by Zaltron et al. (2020). This system initially proposes nine randomly generated faces. Users then have the option to either generate new random faces or mutate faces: Any number of the proposed faces can be locked so that they are not replaced by new random faces. Similarly, any number of faces can be selected for mutation. For this, the average of the selected faces is calculated and new faces are generated by adding random noise to this average face. Participants can control the amount of this noise, i.e., how large the changes should be. Additionally, participants can choose if they want all features to be changed while mutating or only a single random feature. This system can be used to iteratively traverse the latent space until participants decide that one of the proposed faces adequately resembles their mental image. The original method by Zaltron et al. also allowed for explicit manipulation of certain face attributes but given that this requires a dataset with labelled attributes, it is not possible to consider this functionality.

**Control – FaceMaker.** FaceMaker is a tool to manually generate face images using a set of sliders and is used in this work as a control. More specifically, starting with a mean face, participants can manipulate 28 facial features by moving an associated slider. FaceMaker updates the face in real time. Users can adjust these sliders until they are satisfied with the created face resembling their mental image sufficiently well. Since Strohm et al. (2021) used FaceMaker to train the GFE and feature decoder, and it allows for full control over the generation of the faces, we use it as a control condition.

### 4.2. User study

We compared these three methods in a user study with 12 participants (four female) aged between 18 and 29 years (M=23.5, SD=3.6). All participants had normal or corrected-

Table 1: Mean absolute feature distance (MAFD) for different facial regions of our gaze-based method compared to the baseline by Zaltron et al. (2020) and the FaceMaker control condition.

|  | MAFD | | | | |
| Method | Eyes | Nose | Mouth | Jaw | Overall |
| --- | --- | --- | --- | --- | --- |
| **Ours** | **46.3 ± 8** | **44.5 ± 16** | **41.8 ± 12** | **45.7 ± 14** | **40.9 ± 6** |
| Zaltron et al. (2020) | 55.8 ± 13 | 50.6 ± 17 | 54.9 ± 15 | 53.0 ± 22 | 48.0 ± 7 |
| FaceMaker | 30.0 ± 9 | 33.5 ± 11 | 28.9 ± 10 | 33.6 ± 11 | 29.6 ± 6 |

to-normal vision and were recruited through university mailing lists. For our GBC-MIR method, binocular gaze was recorded at 2,000 Hz using a stationary EyeLink 1000 Plus eye tracker. To increase gaze tracking accuracy, we used a chin rest to stabilise participants' heads. Each stimuli covered about 8.9° degrees of visual angle. We counterbalanced the conditions using a within-group Latin square study design.

For each method, participants had to complete three trials in succession. Each of these trials started with a 30 second memorisation step of the target face. After memorisation, participants were instructed to reconstruct the face using the respective method. To prevent memorisation effects of the target faces between methods, each target face was used for each method between groups, not within. After completing three trials of one method they were asked to complete a System Usability Scale (SUS) (Brooke et al., 1996) and NASA-TLX questionnaire (Hart, 2006). Once participants finished all nine trials, they were briefly interviewed about the different systems and finally compensated for their participation.

### 4.3. Reconstruction results

**Metrics.** We report the mean absolute feature distance (MAFD) defined as

$$\text{MAFD} = \frac{1}{28} \sum_{i=1}^{28} |f_i^p - f_i^M|, \tag{2}$$

where $f_i^p$ is the predicted value and $f_i^M$ is the target value for feature $f$. The feature values for $f$ are normalised between 1 and 182 as this is the range given by FaceMaker. Since we calculate the distance between prediction and target feature value a lower MAFD is better. In addition, we report performance for the facial areas *Eyes*, *Nose*, *Mouth* and *Jaw* by grouping features as defined by Strohm et al. (2021).

**Results.** Table 1 shows the performance of our system compared to the baselines. As can be seen, our gaze-based method achieves a MAFD of 40.9 compared to 48.0 by the method of Zaltron et al. (2020), representing an improvement of about 15%. Using the FaceMaker control condition participants achieved an overall MAFD of 29.56. This MAFD is quite

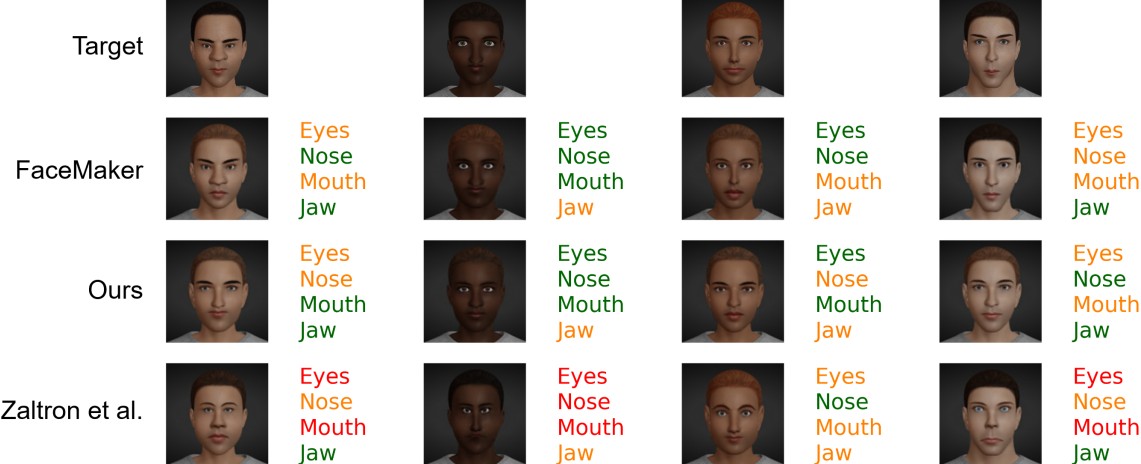

Figure 3: Example reconstructions created during the user study. It shows the reconstructions with our gaze-based collaborative system GBC-MIR (Ours) compared to the two baselines. Colour-coded labels (high, medium, or low) indicate the reconstruction quality of different facial regions. The colour codes represent three equidistant bins between the minimum and maximum MAFD of the test set reconstructions.

considerable given that participants had full and explicit control of the face generation process using FaceMaker, underlining the difficulty of the face reconstruction task.

Figure 3 shows four sample target faces, reconstructions produced by our gaze-based method, as well as baseline reconstructions produced with the method by Zaltron et al. and FaceMaker. Colour-coded feature groups indicate the reconstruction quality of the different facial regions. The colour codes represent three equidistant bins between the minimum and maximum MAFD of the test set reconstructions. We can observe that our method can produce visually plausible mental image reconstructions for most facial regions. Furthermore, as indicated by the MAFD, participants struggle to reconstruct some facial features even when having full control with FaceMaker.

### 4.4. Qualitative evaluation

We conducted an additional 22-participant user study to assess the subjective quality of our reconstructions. Participants were shown sets of four faces, each containing a target face from our test set as well as the reconstructions from the three methods in random order. Participants were asked to rank the three reconstructions from most similar (rank one) to least similar (rank three) according to their similarity with the target. We also asked them to provide text explanations for the reasoning behind their ranking. The average rank given to our reconstructions was $2.28 \pm 0.7$ while the method by Zaltron et al. (2020) ranked $2.40 \pm 0.7$ and the FaceMaker control $1.33 \pm 0.5$.

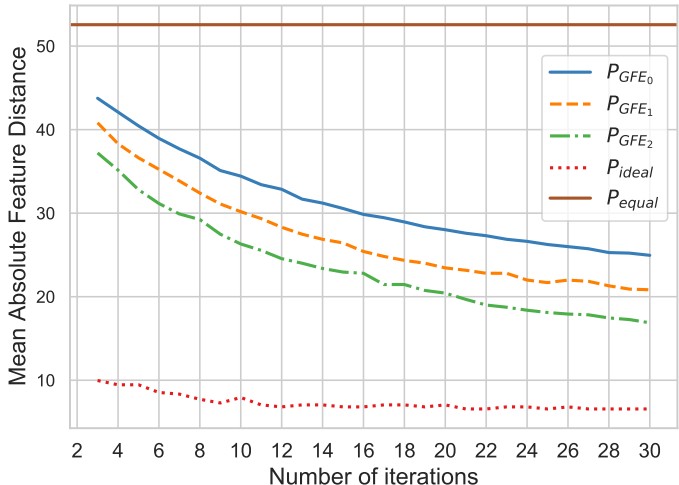

Figure 4: Simulated mean absolute feature distance (MAFD) over number of iterations for different probability distributions of the feature extractor. $P_{\mathrm{GFE}_0}$ are the probabilities estimated for our gaze-guided feature extractors while $P_{\mathrm{GFE}_1}$ and $P_{\mathrm{GFE}_2}$ are better, theoretical versions. $P_{\mathrm{equal}}$ represents a random feature extractor while $P_{\mathrm{ideal}}$ always selects the best feature value possible.

## 5. Analysis and discussion

**Reconstruction quality.**  Our results show that gaze-based mental image reconstruction without prior knowledge about the target is possible and outperforms the previous best manual reconstruction method by ∼15% in MAFD (see Table 1). An additional user study to evaluate the reconstructions underlines this as users tend to rate our reconstructions as more similar to the target image than the reconstructions based on the method by Zaltron et al. (2020). However, analysing the user feedback and comparing qualitative results to the FaceMaker (Schwind et al., 2017) control condition indicate that there is still room for improvement. As faces are perceived holistically (Frowd et al., 2004), a few mismatched features might make a face unrecognisable or, in our case, appear vastly different than the target image. Furthermore, we observed a large variance in the predictions of our system with up to 16 in MAFD for nose related features. This variance can be partly explained by humans' differing ability to memorise images, and in particular faces (Verhallen et al., 2017), as the variance for the FaceMaker control condition is similarly high. This is likely also the reason for the relatively high MAFD of the FaceMaker control method, since participants recreated their mental image of the target, not the target face directly. Finally, we evaluated the SUS questionnaires for each condition. The resulting SUS scores are 52 for the method by Zaltron et al. (2020), 67 for our proposed system and 76 for FaceMaker (see Appendix C for details). Overall these results tend towards a better usability of GBC-MIR compared to Zaltron et al. (2020).

**Feature extractor importance.** Since we simulate the GFE during training of GBC-MIR by replacing it with a selection layer, we are able to analyse the influence of different probability distributions $P_{\mathrm{GFE}}$ on the performance. Figure 4 shows the simulated MAFD for different probability distributions $P$ of the feature extractor over number of iterations. At the extremes, $P_{\mathrm{ideal}}$ assumes a feature extractor always selecting the best possible feature, while $P_{\mathrm{equal}}$ has an equal chance of selecting any feature. The MAFD for $P_{\mathrm{equal}}$ remains at a constant value of 52.56 independent of the number of iterations. Since the system does not receive any information about the target due to the random feature extractor, our network learns to reconstruct the mean face. $P_{\mathrm{GFE}_0}$ shows the simulated MAFD for the probabilities we estimated for the pre-trained GFE used in our system. To analyse how GBC-MIR scales with the quality of the GFE, we simulated results with improved versions $P_{\mathrm{GFE}_{1,2}}$, as shown in Figure 4. For $P_{\mathrm{GFE}_1}$, we subtracted 10% from the probabilities of selecting sub-optimal features (ranks 2-6) and added it to the probability of selecting the best feature (rank 1). In the case of $P_{\mathrm{GFE}_2}$, 20% was redistributed instead. As can be observed in Figure 4, the MAFD of GBC-MIR scales well with the performance of the feature extractor. The MAFD of $P_{\mathrm{GFE}_1}$ is consistently lower over the number of iterations compared to $P_{\mathrm{GFE}_0}$ and analogously between $P_{\mathrm{GFE}_2}$ and $P_{\mathrm{GFE}_1}$. We found that the query engine learns different strategies depending on the GFE performance. For unreliable GFEs it repeatedly samples six equidistant areas in the feature space to determine a value range that is most likely. More reliable GFEs allow to explore the feature space more thoroughly as less repeated measurements to increase confidence are required (see Appendix B for details).

## 6. Conclusion

We have proposed GBC-MIR, the first collaborative system for reconstructing a specific instance of a mental face image from users' eye gaze. In contrast to previous work, our method requires neither sophisticated sensing setup with limited practical relevance as with fMRI, nor any time-consuming and laborious manual image generation process. Through evaluations with a 12-participant eye tracking user study as well as a 22-participant qualitative study, we have shown that our gaze-based method can generate faces that are more similar to the mental target image, and with higher usability than the previous state-of-the-art method. Since our proposed end-to-end collaborative system is independent of any specific image domain, it can be adapted for other image reconstructions tasks. Furthermore, it is even possible to combine our proposed method with pre-trained image feature extractors using human feedback other than gaze information. As such, our results mark an important advance towards collaborative mental image reconstruction and also underline the potential of human gaze for this challenging task.

## 7. Ethical statement

**Data collection.** The data collection procedure was approved by the university's ethics committee and is in line with the General Data Protection Regulation (GDPR) of the European Union. Participants were informed about the task they had to complete and the type of data that was collected. All collected data was fully anonymised and, as such, cannot be traced back to a specific individual, thus ensuring their privacy. The data collection only

started after participants agreed to these conditions and gave their informed consent. They were able to interrupt the study at any time and leave the laboratory. Participants were financially compensated, even if they dropped out early for any reason.

**Potential risks.** Future improvements on this challenging task will likely allow to reconstruct mental images more quickly and with even higher visual fidelity. While they thus have many beneficial applications, such as in police investigations or by helping users with limited expressive skills, involuntary reconstruction poses a non-negligible risk of misuse. However, given that our method still requires users to explicitly search for similar facial features, our current method has little potential for misuse. Nevertheless, it is important to be aware of this potential and to avoid misuse of future methodological developments in this active area of research.

## Acknowledgments

Florian Strohm and Andreas Bulling were funded by the European Research Council (ERC) under the grant agreement 801708. Ekta Sood was funded by the Deutsche Forschungsgemeinschaft (DFG, German Research Foundation) under Germany's Excellence Strategy – EXC 2075 – 390740016. Mihai Bâce was funded by a Swiss National Science Foundation (SNSF) Early Postdoc.Mobility Fellowship.

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

## Appendix A. Evaluating the gaze-guided feature extractor (GFE)

We fine-tuned and evaluated the GFE proposed by Strohm et al. (2021) for mental human-like face reconstruction from FaceMaker (Schwind et al., 2017). In their work they utilised knowledge about the target face to generate sets of six auxiliary faces, ensuring that the target facial features were reflected in the faces shown to the user. In stark contrast, our work does not use prior knowledge, therefore target features are not necessarily reflected in the shown faces. As this might influence the gaze behaviour of users, we collected additional training data to fine-tune their GFE.

### A.1. Data Collection

We recorded gaze data of 7 female and 3 male participants aged between 19 and 35 years (M=27.1, SD=4.7). They were recruited through university mailing lists and compensated for their participation with 15€ per hour. The eyesight of all participants was normal or corrected-to-normal. We recorded their binocular gaze at 2,000Hz using a stationary EyeLink 1000 Plus eye tracker. To increase gaze tracking accuracy, we used a chin rest stabilising participants' head. A 24.4-inch screen with a resolution of $1920 \times 1080$ pixels was placed 90cm in front of the participants to show the face stimuli. Each stimuli covered $8.9°$ degrees of visual angle.

Once participants agreed to our consent form they completed three trials. Each trial started with a calibration-validation procedure to ensure accurate eye tracking. Following this, a target face was shown for 30 seconds that participants had to memorise. After the memorisation phase we iteratively displayed ten sets of six auxiliary faces for 30 seconds each and gave participants the task to rank the faces from one to six according to similarity with their mental image. All face stimuli were generated randomly and independently of each other by sampling the FaceMaker features from a uniform distribution.

### A.2. Fine-tuning and evaluation

We fine-tuned the GFE using 28 out of 30 trials we collected for each participant. Strohm et al. (2021) were able to train their GFE to perform binary classification, since a feature either was the target feature or not. However, since our work does not use prior knowledge, we cannot create binary labels and as such fine-tune their GFE using continuous labels by minimising the mean squared error[1]. The labels are created by calculating the similarity $1 - \text{abs}(f_M - f_i)$ between each feature $f_i \in F_i$ and the target features $f_M \in F_m$.

Using the remaining two trials of each participant we estimated $P_{\text{GFE}}$ by counting how often the model selects the best feature, the second best and so on. Table 2 shows the estimated probability distribution $P_{\text{GFE}}$ for the original model (Strohm et al., 2021) and our model fine-tuned with additional training data and continuous labels. We observe that by adding more task-specific training data, the GFE improved in performance, where the probability of extracting a top-3 feature increases from 61.1% to 68.2%.

---

1. The architecture and all hyper-parameters are equivalent to Strohm et al. (2021).

Table 2: Comparison of $P_{\text{GFE}}$ (i.e., the probability distribution for selecting the k-th best feature value) for the pre-trained model by Strohm et al. (2021) and our fine-tuned model. For our model we observe that the probability of extracting a top-3 feature increases from 61.1% to 68.2%.

| Model | Similarity Rank | | | | | |
|---|---|---|---|---|---|---|
| | 1 | 2 | 3 | 4 | 5 | 6 |
| Strohm et al. (2021) | 18.2% | 20.2% | 22.7% | 16.9% | 12.9% | 9.1% |
| Ours | 22.9% | 21.8% | 23.5% | 16.7% | 10.0% | 5.1% |

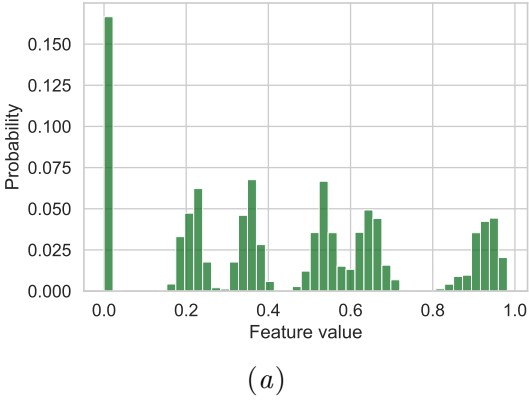

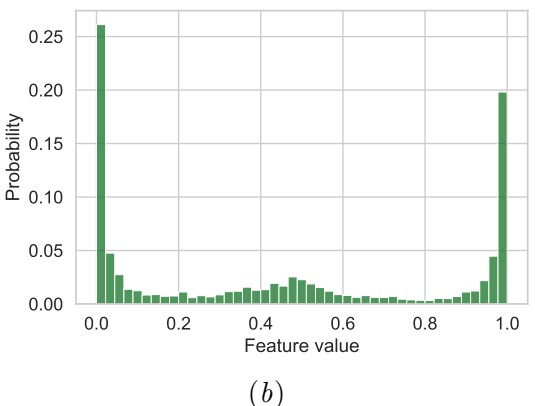

$(a)$ $\qquad\qquad\qquad\qquad\qquad\qquad$ $(b)$

Figure 5: The plots show the probability of query engines to propose a certain auxiliary feature value. (a) shows the feature value distribution for the query engine trained using our estimated values $P_{\text{GFE}_0}$, while (b) shows the feature value distribution for a better estimate $P_{\text{GFE}_2}$.

## Appendix B. Query Engine Performance

To gain a better understanding of how GBC-MIR works, we analyse the behaviour of our query engine. Figure 5 shows the predicted feature value distribution for two ten-iteration query engines trained using $P_{\text{GFE}_0}$ (Figure $5(a)$) and $P_{\text{GFE}_2}$ (Figure $5(b)$). We observe that our query engine in Figure $5(a)$ learned to split the feature space into six roughly equidistant areas, due to the six faces shown. When our query engine is trained with $P_{\text{GFE}_0}$, it learns the strategy to take multiple samples from the same area in order to increase its confidence in a feature's rank, rather than searching the entire feature space for the exact values of the mental image. This can be seen in the higher probability peaks and the surrounding gaps of zero probability. Figure $5(b)$ shows the predicted feature value distribution for our query engine trained with the better simulated GFE $P_{\text{GFE}_2}$. We observe a more uniform distribution of the predictions across the feature space, as there are no longer

any surrounding gaps of zero probabilities. With higher confidence in the selected feature's rank, the query engine does not require redundant samples and can use a more fine-grained search strategy.

## Appendix C. Usability and Workload

**System Usability Scale**    We asked participants to fill in a System Usability Scale (SUS) questionnaire (Brooke et al., 1996) to assess the usability of our three conditions. The SUS allows us to calculate a single score on a scale from 0-100 (higher is better), by answering ten standardised questions on a five point Likert scale. The resulting SUS scores are 52 for the method by Zaltron et al. (2020), 67 for our proposed system and 76 for FaceMaker. A repeated measures one-way ANOVA test (Gueorguieva and Krystal, 2004) indicated a significant difference (p=0.007) between the SUS scores of the three conditions. Pairwise Tukey's HSD post-hoc tests (Abdi and Williams, 2010) indicate a significant difference only between the method by Zaltron et al. (2020) and FaceMaker (p=0.026). Overall these results tend towards a better usability of GBC-MIR compared to Zaltron et al..

**NASA-TLX Workload**    Additionally to the SUS, we asked participants to complete the Raw-NASA-TLX questionnaire (Hart, 2006), allowing us to assess the perceived workload of the participants for each condition on a scale from 1-100 (lower is better). The resulting values are 46 for the method by Zaltron et al. (2020), 47 for our proposed system and 30 for FaceMaker. A repeated measures one-way ANOVA test (Gueorguieva and Krystal, 2004) indicate a significant difference (p=0.002) between the TLX values of the three conditions. Pairwise Tukey's HSD post-hoc tests (Abdi and Williams, 2010) indicate a significant difference between the method by Zaltron et al. and FaceMaker (p=0.022) as well as between our method and FaceMaker (p=0.015). Overall, these results suggest that the usage of FaceMaker was the least demanding task, while there is no significant difference between GBC-MIR and the method of Zaltron et al..

