# OpenReview forum: "Facial Composite Generation with Iterative Human Feedback"
_NeurIPS.cc/2022/Workshop/GMML — Gaze Meets ML 2022 Oral_

### Official Review · Reviewer_VRWv · 2022-10-15
**Recommending acceptance**

**Rating:** 8
**Confidence:** 4

**Review:**

This paper proposed a human-AI collaborative system to reconstruct the user's mental image about another person's face using the eye gaze data. The proposed system does not need prior knowledge of the target face and only rely on implicit behavioral cues. The contribution and novelty is clearly stated. I would recommend acceptance of this paper.

Here are some comments for the authors:
1. Fig. 2: the vertical text is not reader-friendly. Please consider updating this diagram with horizontal text.
2. Line 238-239: the authors can elaborate a little bit more on the target reconstruction and how cross-contamination of the target reconstruction is avoided.
3. For Fig. 3, it is not clear where the color-coded labels come from. Is it from the user qualitative study?
4. For qualitative study, the authors show the average rank. What is the standard deviation of the ranking score?

---

### Official Review · Reviewer_xWFz · 2022-10-15
**The authors propose a novel way to reconstruct facial images using human-AI interactions and learnable recurrent units. However, the proposed architecture is unclear, and the methodology is hard to follow.**

**Rating:** 5
**Confidence:** 3

**Review:**

**Strengths:**

* The paper is well-motivated, and the authors have provided relevant information about the prior works.

* As a novel contribution this model uses the recurrent units to improve the generation of auxiliary images without human intervention iteratively and finally generate an image close to the mental image.

* One of the main goals is to use gaze data to identify important features; however, it is impossible while training end-to-end. The authors have identified a novel feature selection strategy that mimics graze-based feature selection.
* In a comparative study, the authors have provided evidence that their approach performs better than the baseline.

**Weakness:**

The methodology is unclear. It's hard to understand how the authors have trained the model, and there are several inconsistencies in the information flow through the model.

* In Figure 1, an important module is the query engine. However, the authors did not specify how they implemented that in the methodology section. Is it done using the GRUs?

* The algorithm for the selection layer is hard to understand. If I understand correctly, $f_{I,j}$ is a matrix, so in Algorithm 1, $differences$ should be a matrix of dimension $No.Features \times No.OfAuxilliaryImages$. Using this $differences$ matrix, they are creating a $P_{GFE}$ where every row corresponds to the features, and every column gives the probability of that auxiliary image being similar to the mental image. In the algorithm, I don't understand how they are sampling a single number $k$ from the matrix $P_{GFE}$ and using them for all the features.

* The information flow through the projection layer, and the GRU is difficult to follow. The authors have mentioned in the projection layer, the 28 features from 6 auxiliary images are combined into a single feature of dimension 50. This feature is passed through 50 GRUs. GRUs are recurrent modules whose inputs are sequential features across time. In this model, it's unclear how they generate 50 sequential features when the whole module is iterating only $m = 10$ times.

* In the paper, the authors wrote,
	"The outputs of each recurrent layer from the two paths, selection, and projection, are concatenated and input to the next module of iteration j + 1." After the concatenation, it is unclear how they generate the new set of auxiliary images for the next iteration. Additionally, it would be helpful if the authors could provide the dimensions of the features after selection and projection and explain how they are modified to new images.


* The quantitative performance shows a better mean compared to the baseline. However, the standard deviation is high. So, it would be helpful if the authors could do a statistical test and provide the p-value.

---

### Official Review · Reviewer_oJsB · 2022-10-18
**Use gaze detection to facilitate faster construction of human faces.**

**Rating:** 8
**Confidence:** 3

**Review:**

I very nice application of gaze tracking to facilitate faster construction of faces in various environments; e.g., police sketches or video game avatars. Good experimental design, good baseline, good results.

The application of a usability followup was a nice touch. While you present a single approach per reconstruction, one wonders if getting in the "ballpark" with gaze based construction, then handing the user FaceMaker to do final tuning would work.


Nit:
* Line 89 - require -> requires

---

### Meta-Review · Area_Chair_8zuP · 2022-10-20

**Recommendation:** Accept (Oral)
**Confidence:** 4

**Metareview:**

The authors propose an interesting research work related to reconstruction of mental images based on human-AI collaboration. They show that the gaze-based method can generate better results without the need of manual annotations. The paper is organized very well and they provide comparisons with baseline methods. Reviewers seem to agree on the novelty and contributions but it is highly recommended that they address questions from reviewers regarding clarity of the methodology. If these concerns are addressed the quality of the work would be further improved.

---

### Decision · Program_Chairs · 2022-10-20

Accept (Oral)